# The Digital Divide: A Retrospective Survey of Digital Rectal Examinations during the Workup of Rectal Cancers

**DOI:** 10.3390/healthcare9070855

**Published:** 2021-07-06

**Authors:** Omar Farooq, Ameer Farooq, Sunita Ghosh, Raza Qadri, Tanner Steed, Mitch Quinton, Nawaid Usmani

**Affiliations:** 1Division of General Surgery, Department of Surgery, University of Alberta, Edmonton, AB T6G 2R3, Canada; ofarooq@ualberta.ca (O.F.); rqadri123@gmail.com (R.Q.); mitch.quinton@gmail.com (M.Q.); 2Section of Colorectal Surgery, Department of Surgery, University of British Columbia, Vancouver, BC V6T 1Z4, Canada; 3Department of Oncology, University of Alberta, Edmonton, AB T6G 1Z2, Canada; sunita.ghosh@albertahealthservices.ca (S.G.); tsteed@ualberta.ca (T.S.)

**Keywords:** digital rectal examination (DRE), rectal cancer, primary care, physical examination, rectal bleeding

## Abstract

*Background:* Digital rectal examination (DRE) is considered an important part of the physical examination. However, it is unclear how many patients have a DRE performed at the primary care level in the work-up of rectal cancer, and if the absence of a DRE causes a delay to consultation with a specialist. *Methods:* A retrospective patient questionnaire was sent to 1000 consecutive patients with stage II or stage III rectal cancer. The questionnaire asked patients to recall if they had a DRE performed by their general practitioner (GP) when they first presented with symptoms or a positive FIT test. Demographic data, staging data, and time to consultation with a specialist were also collected. *Results:* A thousand surveys were mailed out, and a total of 262 patients responded. Of the respondents, 46.2% did not recall undergoing a digital rectal examination by their primary care provider. Women were less likely to undergo a DRE than men (28.6% vs. 44.3%, *p* = 0.019). While there was a trend towards longer times to specialist consultation in patients who did not undergo a DRE (27.0 vs. 12.2 weeks), this was not statistically significant (*p* = 0.121). *Conclusion:* A significant proportion of patients who are FIT positive or have symptomatic rectal bleeding do not recall having a DRE by their primary care provider. Barriers may include lack of comfort with performing DRE or lack of time. Clearer guidelines and more support for GP’s may increase uptake of DRE.

## 1. Introduction

Colorectal cancer is the second most commonly diagnosed cancer in men and women, and second leading cause of cancer death [1]. The digital rectal examination (DRE) is a key physical exam maneuver in the detection and diagnosis of rectal cancers. As the surgical adage goes, “The only reasons for not performing a DRE is the lack of an anus or the lack of an examining finger.” For the surgeon, the DRE is a key maneuver in surgical planning to help determine the position of the rectal cancer, its relationship to the sphincter complex, and the nature of the tumor (fixed or mobile) [2]. A DRE is considered part of the standard of care in the evaluation of a patient with anorectal complaints or rectal bleeding according to both provincial and national guidelines [3,4].

However, it is becoming less clear what value the DRE has at the level of the primary care physician. Several studies have challenged the sensitivity and specificity of the DRE [5,6,7,8,9]. These studies are balanced against the potential delays that might result from not performing a DRE. Given that many endoscopists and surgeons triage patients based on narrative comments in referrals, the description of a “palpable mass” on a referral would prompt gastroenterologists and surgeons to expedite the work-up or consultation appointments of these patients.

In Northern Alberta, all patients with stage II and higher rectal cancer are seen at the Cross Cancer Institute. This referral process allows for centralized review of all cases, ensuring that appropriate radiation oncology and surgical consultations are made. In this study, we surveyed patients to determine the rate of DREs performed by general practitioners (GPs) on rectal cancer patients. We then evaluated the impact of the DRE by GPs on wait times for specialist evaluation.

## 2. Methods

### 2.1. Setting 

Cross Cancer Institute, a large, multidisciplinary, cancer clinic in Edmonton, Alberta. The Cross Cancer Institute receives complex referrals from throughout Northern Alberta as well as the Northwest Territories.

### 2.2. Inclusion/Exclusion Criteria

We identified consecutive patients seen at the Cross Cancer Institute in Edmonton, Alberta with a pathological diagnosis of stage II or stage III rectal cancer. Patients were included if they were symptomatic at time of presentation or if they had a positive FIT test. Stage I patients were excluded as these patients are not routinely referred to the Cross Cancer Institute for consideration of neoadjuvant or adjuvant treatment.

### 2.3. Study Design

We used the Alberta Cancer Registry to identify consecutive patients with stage II or stage III rectal cancer patients in Northern Alberta from January 2013 until December 2017. Identified patients had a questionnaire mailed out to them that included a consent form for enrolment in the study. The questionnaire collected information on the dates of their initial assessments, whether a DRE was performed, and a general reflection of the DRE if done (see Appendix A for the detailed questionnaire used). The mail-out included pre-stamped return envelopes to help improve the response rates of returned questionnaires.

The primary objective of the study was to determine the proportion of patients who recalled undergoing a digital rectal examination by their primary care physician. The secondary objective was to determine the time to consultation from their initial visit with their primary care physician to the Cross Cancer Institute.

The sample size calculations determined that about 106 patients who underwent a DRE and 106 patients who did not undergo a DRE would be able to detect a 19% difference with a power of 80% and a significance level of 5%. Hence, a sample size of 212 subjects was estimated. Assuming about 25% missing data, we would need an additional 53 patients to account for the missing data. As a result, a total of 265 patients (132 patients in DRE and 133 in non-DRE arm) was estimated as a minimum sample size. The sample size calculations were performed using a two-sided Mann–Whitney test assuming that actual distribution is normal.

### 2.4. Statistics

For the data analysis, anonymized patient information was used to maintain data confidentiality. Basic demographic information for the patients were collected from the Alberta Cancer Registry (ACR). This registry centrally collects basic information about all cancer patients in Alberta, allowing researchers to extract key demographics and pathology, such as date of birth, stage, site of tumor, histology, and postal code. The primary outcome was the percentage of patients who recalled having a DRE prior to presentation to a specialist (defined as a medical or radiation oncologist, surgeon, or gastroenterologist). The secondary outcome was wait-time from first presentation to primary care provider to consultation with specialist.

Descriptive statistics were used to represent the study variables, with mean, median, and standard deviations reported for continuous variables. Frequency and proportions were reported for categorical variables. One-way ANOVA was used to compare the mean of the continuous variables between the two study arms. Chi-square tests were used to study the association between the two categorical variables. A *p*-value < 0.05 was used for statistical comparisons and two sided tests were used. SAS (SAS Institute Inc., Cary, NC 27513, USA) version 9.3 software was used to analyze the data. A *p*-value < 0.05 was used for statistical significance and two-sided tests were used.

## 3. Results

### 3.1. Patient Population

Of the 1000 questionnaires mailed out, 262 (26.2%) were completed and returned by patients. The baseline characteristics for patients are shown in Table 1. There was a preponderance of male patients and stage III patients in those that responded. Unfortunately, demographic and tumor data for 73 patients could not be extracted from the returned questionnaires, due to an inability to identify patients from the returned questionnaires in these patients.

The majority of patients (144, 55%) recalled presenting to their GP with rectal bleeding and 104 (39.7%) with a positive FIT test (Table 1). Interestingly, 48 out of 262 patients (18.3%) recalled getting a FIT test, despite having symptomatic or obvious rectal bleeding.

### 3.2. Patients Undergoing DRE

The majority of patients referred to the CCI did not recall having a DRE performed by their GP prior to seeing a specialist (Table 2). Out of the cohort of 262 patients, 121 (46.2%) patients did not recall having a DRE, 96 (36.6%) had a DRE, with 27 (10.3%) of patients unsure if they had a DRE (*p* = 0.03). Patients who presented to their GP with rectal bleeding were more likely to recall having had a DRE; 46 (17.5%) patients with rectal bleeding reporting having had a DRE, while only 29 (11.0%) patients who had did not have rectal bleeding underwent a DRE (*p* = 0.684). About 35 (33.7%) patients who had positive FIT test also had a DRE done, while 55 (52.9%) FIT test positive patients did not have DRE done. This association was not statistically significant (*p* = 0.301). About 15 (5.7%) did not have any recollection of rectal bleeding, a FIT test, or DRE being performed. Female patients were less likely to have a DRE, with only 18 (28.6%) females recalling to have had a DRE, while 44 (47.3%) of the males had DRE. This difference was statistically significant (*p* = 0.019).

### 3.3. Secondary Outcomes

The wait time for all patients to see a specialist was 95 days (range: 0 to 735 days). Having a DRE performed by the GP trended towards a shorter wait time to being seen at the CCI, with a mean wait time of 12.2 weeks versus 27.0 weeks (Table 3). Although this trended towards shorter wait times, this was not statistically significant (*p*-value 0.121).

## 4. Discussion

Rectal cancer continues to be a major source of cancer-related morbidity and mortality across North America [1]. While there have been significant advances in the treatment of rectal cancer, including watch-and-wait, total neoadjuvant therapy, as well as plethora of surgical techniques, the fundamental approach to rectal cancer continues to rely on prompt diagnosis [2]. The DRE remains important for the surgeon as it gives key information regarding tumor height, location, mobility, and relation to anorectal ring [2]. A study by Tanaka and colleagues found that the DRE, flexible sigmoidoscopy, and rigid proctoscopy obtained highly concordant values for the height and location of the tumor [10]. The DRE is also important for triaging patients, as a palpable rectal mass would prompt most surgeons and endoscopists to expedite appointments and/or endoscopy. There are no established guidelines in Canada for the expected wait time to see a specialist for a patient with a palpable rectal lesion. However, at the CCI, a referral with the presenting complaint of “palpable rectal mass” prompts consultation within four weeks. The impetus of this study was to identify the percentage of patients who underwent a DRE prior to being seen by a specialist, as anecdotally it seemed that the majority of patients referred for a potential rectal cancer were not undergoing a DRE.

In this study, we found that 36.6% of patients recalled undergoing a digital rectal examination by their GP prior to being seen by a specialist. Analysis of our secondary outcomes found that not having had a DRE trended towards longer wait times to see a specialist, but did not reach statistical significance. Women were also less likely to have a DRE on presentation than their male counterparts (71.4% vs. 52.8%, *p* = 0.014).

The value of the DRE performed by GPs as a diagnostic tool has been challenged. While the DRE has been emphasized as an important physical examination maneuver, it has poor sensitivity as a diagnostic tool. Ang and colleagues found that DRE performed by GPs had a sensitivity of 0.76 and positive predictive value of 0.296 for rectal lesions. It had a reasonably high specificity at 0.917 [5,6]. This is echoed by the findings in the prostate cancer literature, with the DRE found to be a poor screening tool [11]. In addition, FIT testing is ubiquitously available in Alberta. In our study, we found that a significant percentage of patients with symptomatic rectal bleeding were also having FIT tests performed. In other words, GP’s may be using FIT tests in place of a DRE for the detection of colorectal lesions. All these factors may contribute to the low percentage of patients undergoing a DRE. Our findings echo findings from other investigators. In a study done in Boston, only 22% of providers performed an adequate physical exam when working up a patient with rectal bleeding [12].

It is also worth noting that women were less likely to recall having had a DRE than men. This may again highlight the issues that may be at play with respect to privacy and patient comfort. GPs may have limited experience, education, and comfort with performing a DRE. Finally, GPs face considerable time pressure while seeing these patients, and may not have adequate space or chaperones to be able to perform a DRE. Our work echoes that of Hennigan and colleagues, who found that barriers to performing DRE’s for GPs included reluctance of the patient, expectation that the examination would be repeated, lack of a chaperone, and lack of time [13].

Our study demonstrated a trend towards longer wait times if patients did not undergo a DRE. While we were unable to demonstrate a difference in outcomes for patients who did not undergo a DRE, a number of studies have demonstrated that delays in treatment for colorectal cancer patients increase health care costs [14,15]. There is mixed data on whether delays in diagnosis impacts outcomes [15,16]. Further work should focus on quantifying the impact of inappropriate or incomplete workup of rectal cancer on outcomes, particularly in local contexts.

There are important implications from our study. Focus should be placed on educating GPs on the correct technique for performing a DRE. In a survey performed by Bussieres and colleagues, 33% of family medicine residents stated they had never received practical teaching in or supervision of DRE technique during medical training [17]. More than half of the participants considered their training to be average or insufficient [17]. Education around DRE may thus improve the confidence of primary care physicians to perform DREs [18]. Normalizing the presence of chaperones during physical exam maneuvers for every patient may help to increase the rate of DRE performance. There also may be innovative solutions to lessen the burden on already taxed GPs. The development of a specialized, centralized anorectal clinic staffed by interested GPs or even nurse-practitioners can help to work up this patient population more thoroughly. Further investigation should be done to process map the diagnosis of rectal cancer in Alberta, similar to process mapping performed for other general surgical conditions [19,20,21]. In a post-COVID19 world, it is critical to identify key patient presentations that should mandate an in-person consultation (versus telehealth) and performance of physical examination maneuvers such as a DRE [22].

There are several limitations to this study. Firstly, we utilized a retrospective questionnaire due to the difficulties in obtaining the referral letters and clinical notes from referring GPs. While most patients were able to remember if they had undergone a DRE, the results are subject to recall bias. Other than asking patients if they recall having had a DRE, it is difficult to capture this data objectively. Physicians often poorly document their findings, as has been found in studies across a variety of specialties and settings [23,24,25]. A chart review, therefore, would not necessarily provide more reliable data. Secondly, we were unable to capture the overall number of patients who are undergoing DREs by their GPs and perhaps appropriately being sent to a benign anorectal specialist or other specialist. Unfortunately, complete demographic and tumor information was not available for all of the patients in this study, limiting some of the conclusions and introducing a source of potential bias. In addition, a sample size of 265 patients had been calculated to achieve statistical significance. We felt that 262 responses was quite close to that number and an additional three participants was unlikely to change our overall conclusions. Finally, we had a low response rate of approximately 26.2%. While there was no significant difference between responders and non-responders from a tumor or demographic perspective, our results are subject to response bias.

## 5. Conclusions

The DRE is an underutilized diagnostic tool by GPs and can contribute to delays in definitive care in rectal cancer patients. Additional work should be done to examine contributing factors to underutilization of DRE and methods to mitigate these challenges.

## Figures and Tables

**Table 1 healthcare-09-00855-t001:** Patient characteristics (age, sex, tumor characteristics, FIT positive, percent with rectal bleeding). Statistically significant differences in proportions are bolded (*p* < 0.05). *p*-values were calculated using appropriate univariate statistics.

Variable	Results	*p*-Value
Sex	Male: 112 (42.7%) Female: 77 (29.4%) Unknown: 73 (27.9%)	**0.0015**
Patient Location	Rural: 29 (10.7%) Urban: 233 (89.3%)	**<0.0001**
Rectal Bleeding	Yes: 144 (55.0%) No: 80 (30.5%) Unsure: 20 (7.6%) Missing: 18 (6.9%)	**<0.0001**
DRE Performed	Yes: 96 (36.6%) No: 121 (46.2%) Unsure: 27 (10.3%) Missing: 18 (6.9%)	**0.03**
FIT Positive	Yes: 104 (39.7%) No: 113 (43.1%) Unsure: 27 (10.3%) Missing: 18 (6.9)	0.5
Number of Patients with Bleeding and FIT Positive	48 (18.3%)	0.05
Pathological Staging	Stage IIA: 47 (17.9%) Stage IIB: 2 (0.8%) Stage IIC: 1 (0.4%) Stage IIIA: 22 (8.4%) Stage IIIB: 98 (37.4%) Stage IIIC: 13(5.0%) Stage III NOS: 6 (2.3%) Unknown: 73 (27.9%)	0.05

**Table 2 healthcare-09-00855-t002:** Number of patients who recalled having had a DRE.

	Presenting with Rectal Bleeding	Presented with Positive FIT Test	Gender
Recall having a DRE by GP	59 (41.0%)	35 (33.7%)	Female: 18 (24.7%) Male: 45 (40.1%)
Do not Recall Having a DRE by GP	72 (50.0%)	55 (52.9%)	Female: 47 (61.6%) Male: 56 (50%)
Unsure	13 (9.0%)	14 (13.5%)	Female: 10 (13.7%) Male: 16 (14.3%)
Total	144	104	262

**Table 3 healthcare-09-00855-t003:** Self-reported time between being seen by GP and specialist (gastroenterologist, surgeon, or oncologist).

Variable	Mean Wait Times in Weeks (Standard Deviation)	*p*-Value
Rectal bleeding—Yes	25.0 (± 64.6)	
Rectal Bleeding—No	12.5 (± 18.3)	0.202
FIT test—Positive	17.7 (± 42.6)	
FIT test—Negative	22.0 (± 60.1)	0.659
DRE Performed by GP	12.2 (± 25.0)	
No DRE Performed by GP	27.0 (± 67.8)	0.121

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
