# Peer review of "The Digital Divide: A Retrospective Survey of Digital Rectal Examinations during the Workup of Rectal Cancers"

_healthcare, 2021, doi:10.3390/healthcare9070855_

Round 1
Reviewer 1 Report
Rectal cancer displays a major health burden and metastasis and disease recurrence remain challenging. Digital rectal examination (DRE) is a key approach in clinical practice and, therefore, the authors performed a survey among stage II or III rectal cancer patients. Their findings on this topic indicate gender differences in patients undergoing DRE and will be of interest to the readership of Healthcare. A number of minor points should be addressed prior to publication as detailed below.
The sentence “A retrospective patient questionnaire was sent to 1000 consecutive patients with stage II or stage II rectal cancer.” (abstract) should be changed to “[…] stage II or stage III […]”.
Please add p-values to the clinical parameters presented in Table 1.
Since the authors included a high percentage of patients with unknown gender into their study, the parameter “male” should be added to Table 2. Otherwise, the reader will not know to what extent the remaining percentage is male or unknown.
Reviewer 2 Report
Dear authors,
I really like your research topic, but I think it could be improved with a prospective design. A new cohort of patients and General Practitioners could improve your research plan.
I miss some data at sample size calculation (based on official registrations), some flaws in the abstract, like not showing the final sample of 262 patients, with some issues at data recording process (like gender or diagnosis, 28% loss in gender and 73 diagnosis and demographic data were missing).
No significant results could change with a larger sample.
Round 2
Reviewer 2 Report
My dear authors,
Although your research topic is very interesting and we can consider that it has adequate internal validity, I believe that it is not sufficiently relevant to be published in Healthcare.
Best regards
Author Response
Thank you for the comments. We do believe that this is an important topic for the journal of Healthcare to publish, especially in a COVID 19 era. With the increasing prevalence of televisits and virtual consults, it is important to revisit the relevance of physical exam maneuvers such as the digital rectal examination. We showed that even before COVID19, only about half of rectal cancer patients recalled having a DRE performed. Furthermore, female patients in particular were less likely to have had a DRE performed. Our manuscript highlights this important topic and will likely find a broad audience among primary care physicians, surgeons, and oncologists.
Thank you again for your consideration.